# Disseminated *Mycobacterium genavense* Infection Mimicking Sarcoidosis: A Case Report and Review of Literature on Japanese Patients

**DOI:** 10.3390/microorganisms11092145

**Published:** 2023-08-24

**Authors:** Ryo Ogata, Takashi Kido, Kazuaki Takeda, Kazuki Nemoto, Riko Heima, Mami Takao, Ritsuko Miyashita, Mutsumi Ozasa, Takatomo Tokito, Daisuke Okuno, Yuya Ito, Hirokazu Yura, Tomohiro Koga, Kunio Hashimoto, Shinnosuke Takemoto, Takahiro Takazono, Hiroshi Ishimoto, Noriho Sakamoto, Kazumasa Fukuda, Yuka Sasaki, Yasushi Obase, Yuji Ishimatsu, Kazuhiro Yatera, Koichi Izumikawa, Hiroshi Mukae

**Affiliations:** 1Department of Respiratory Medicine, Nagasaki University Hospital, Nagasaki 852-8501, Japan; dositano888@gmail.com (R.O.); k-takeda@nagasaki-u.ac.jp (K.T.); r.miya@nagasaki-u.ac.jp (R.M.); 0717mutumi@gmail.com (M.O.); t-takatomo@nagasaki-u.ac.jp (T.T.); vkvkv10101@gmail.com (D.O.); y.ito@nagasaki-u.ac.jp (Y.I.); h-yura@nagasaki-u.ac.jp (H.Y.); shinnosuke-takemoto@nagasaki-u.ac.jp (S.T.); takahiro-takazono@nagasaki-u.ac.jp (T.T.); h-ishimoto@nagasaki-u.ac.jp (H.I.); nsakamot@nagasaki-u.ac.jp (N.S.); obaseya@nagasaki-u.ac.jp (Y.O.); hmukae@nagasaki-u.ac.jp (H.M.); 2Department of Respiratory Medicine, University of Occupational and Environmental Health, Japan, Kitakyusyu 807-8556, Japan; kazuki585@outlook.jp (K.N.); yatera@med.uoeh-u.ac.jp (K.Y.); 3Clinical Genomics Center, Nagasaki University Hospital, Nagasaki 852-8501, Japan; riko-h@nagasaki-u.ac.jp (R.H.); takaomami@nagasaki-u.ac.jp (M.T.); 4Department of Immunology and Rheumatology, Division of Advanced Preventive Medical Sciences, Nagasaki University Graduate School of Biomedical Sciences, Nagasaki 852-8501, Japan; tkoga@nagasaki-u.ac.jp; 5Department of Pediatrics, Nagasaki University Graduate School of Biomedical Sciences, Nagasaki 852-8501, Japan; kunioh@nagasaki-u.ac.jp; 6Department of Infectious Diseases, Nagasaki University Graduate School of Biomedical Sciences, Nagasaki 852-8501, Japan; koizumik@nagasaki-u.ac.jp; 7Department of Microbiology, University of Occupational and Environmental Health, Japan, Kitakyusyu 807-8556, Japan; kfukuda@med.uoeh-u.ac.jp; 8Center for Pulmonary Diseases, National Hospital Organization Tokyo National Hospital, Tokyo 204-8585, Japan; sasakiy2012@gmail.com; 9Department of Nursing, Nagasaki University Graduate School of Biomedical Sciences, Nagasaki 852-8520, Japan; yuji-i@nagasaki-u.ac.jp

**Keywords:** *Mycobacterium genavense*, sarcoidosis, systemic inflammatory diseases

## Abstract

Sarcoidosis is a systemic inflammatory disease characterized by noncaseating epithelioid cell granulomas. However, certain infections can exhibit similar histological findings. We present a case of a 69-year-old man who was initially diagnosed with sarcoidosis and later was confirmed, through 16S rRNA sequencing, to have disseminated *Mycobacterium genavense* infection. Acid-fast bacteria were detected in the bone marrow biopsy using Ziehl–Neelsen staining, but routine clinical tests did not provide a definitive diagnosis. The patient tested negative for HIV, anti-interferon-gamma antibodies, and genetic immunodeficiency disorders. He was treated with multiple drugs, including aminoglycosides and macrolides, but showed no improvement in fever and pancytopenia. However, these clinical signs responded favorably to steroid therapy. We reviewed 17 Japanese cases of *M. genavense* infection. All cases were in males; 7/17 (41%) were HIV-negative; and 12/17 (71%) had a decreased CD4 count. Genetic analysis confirmed *M. genavense* isolation, and macrolides were used universally. *Mycobacterium genavense* infection is challenging to identify and mimics other systemic inflammatory diseases such as sarcoidosis. There are no standard treatment protocols. Our case report and Japanese case review contribute to understanding this rare disease.

## 1. Introduction

Sarcoidosis is a systemic inflammatory disease of unknown etiology, characterized by noncaseating epithelioid cell granulomas in the involved organs. In addition to the presence of histopathological noncaseating epithelioid cell granulomas, an increased lymphocyte count and CD4/8 ratio in bronchoalveolar lavage fluid (BALF), and elevated serum angiotensin-converting enzyme (ACE) levels, can provide valuable clues to the diagnosis. However, sarcoidosis must be differentiated from sarcoid-like lesions secondary to infections, malignancy, or other causes [1]. Differentiation from infection can be challenging, especially when pathogens are rare and difficult to detect. As a therapeutic approach, prednisolone (PSL) is often initiated at approximately 0.5 mg/kg/day. The dose is gradually reduced to 5–10 mg/ day for maintenance treatment in order to prevent relapses and reduce the risk of complications.

*Mycobacterium genavense* is a nontuberculous mycobacterium that was first reported in 1992 to cause disseminated infection in *human immunodeficiency virus* (HIV) patients [2]. It also causes disseminated infections in organ transplant recipients and in patients receiving immunosuppressive therapy for autoimmune diseases [3]. Owing to the difficulty of the detection of organisms, identification of the pathogen is often challenging. Thus, cases of *M. genavense* infection that were initially diagnosed as sarcoidosis or other systemic inflammatory diseases have been reported [1,4]. Here, we present a patient with *M. genavense* infection who was initially diagnosed with sarcoidosis. We also reviewed Japanese cases of *M. genavense* infection described in Japanese and English to gather evidence and explore the characteristics of this rare disease.

## 2. Case Presentation and Review of Literature on Japanese Patients

A 65-year-old Japanese man with fever and pancytopenia was referred to our hospital. On admission, his vital signs were as follows: body temperature, 37.1 °C; pulse rate, 85 beats/min; and respiratory rate, 16 breaths/min. The patient’s respiratory sounds were normal, but he had erythema nodosum on both feet. Hematology revealed a reduced white blood cell (WBC) count of 2000/µL, with 62.6% neutrophils, 29.7% lymphocytes, 6.0% monocytes, and 0.5% eosinophils; a hemoglobin (Hb) level of 10.9 g/dL; and a platelet (PLT) count of 85,000/μL. Blood biochemistry revealed elevated levels of C-reactive protein (CRP) (1.82 mg/dL), ACE (30.5 U/L), soluble interleukin-2 receptor (sIL-2R) (2381 U/mL), and Krebs von den Lungen-6 (KL-6) (882 U/mL). Anti-glycopeptidolipid (GPL)-core IgA antibody for detecting *Mycobacterium avium* complex (MAC) was positive (1.11 U/mL), whereas the interferon-gamma release assay (IGRA) was negative. Representative test results for collagen disease-related autoantibodies were negative.

High-resolution computed tomography (CT) revealed a reticular shadow and thickening of the intralobular septa in the lower zones of both lungs (Figure 1). No mediastinal lymphadenopathy was observed. A bronchoscopy was also performed, and BALF obtained from the right B^5^ yielded 3.0 × 10^5^ cells/mL (19% macrophages, 78% lymphocytes, 3% neutrophils, and 0% eosinophils) with a high CD4/CD8 ratio of 5.3. Ziehl–Neelsen staining, polymerase chain reaction (PCR) for *M. tuberculosis* and MAC, and BALF culture for *Mycobacterium* species were negative. Pathological examination of specimens obtained using transbronchial lung biopsy through the right B^2^b, B^4^a, and B^8^a bronchi and skin biopsies showed noncaseating granulomas. Bone marrow biopsy also revealed noncaseating granulomas with hypocellular marrow. Ziehl–Neelsen staining and all specimens were negative for *Mycobacterium* on PCR testing. The patient was accordingly diagnosed with sarcoidosis based on the increased levels of serum ACE, proportion of lymphocytes, CD4/8 ratio in the BALF, and noncaseating granulomas with negative Ziehl–Neelsen staining. PSL 30 mg/day (0.5 mg/kg/day) was initiated. As the patient’s fever improved, his WBC count recovered to a level above 5000/µL, with lymphocytes comprising approximately 25%. His Hb level also increased to >14 g/dL, and the PLT count reached approximately 150,000/μL. The PSL dose was gradually reduced to a maintenance dose of 12.5 mg per day.

Four years later, at the age of 69 years, the patient developed a persistent fever. Hematology revealed a decreased WBC count at 2600 /µL, with lymphocytes accounting for 8.9%. His hemoglobin count was 10.4 g/dL, and his PLT count was 76,000 /μL. Ziehl–Neelsen staining of the pathological specimens obtained by bone marrow aspiration revealed a noncaseating granuloma (Figure 2A,B), although PCR for tuberculosis and MAC was negative. The results of acid-fast bacilli culture were also negative. We performed 16S ribosomal RNA gene sequencing of *Mycobacterium* species, as described previously [5]. Although a previous study identified *Mycobacterium species* using an average 550 bp nucleotide sequence, in this study, we improved the identification accuracy further by using a longer nucleotide sequence of 973 bp. Finally, the patient was diagnosed with *M. genavense* infection. However, *M. genavense* was not isolated from the sputum, BALF, blood, urine, or stools. Thus, in order to examine drug sensitivity, we attempted to culture acid-resistant bacteria from a re-biopsied bone marrow sample using mycobactin J-containing medium [6]. The culture attempt was unsuccessful, possibly because of the effects of the treatment. Fluorodeoxyglucose positron emission tomography showed abnormal uptake in the bone marrow, particularly in the femur and pelvis.

Although the patient had no history of recurrent opportunistic infections since childhood, his immunological status was re-evaluated. The WBC and lymphocyte counts were 1200/µL and 370/µL, respectively. The proportion of CD4+ T lymphocytes was 64/µL (17.2%), and that of CD3+ and CD19+ T lymphocytes was 74.2% and 0.2%, respectively. Immunoglobulin levels were as follows: IgG, 858 mg/dL; IgA, 112 mg/dL; and IgM 36.4 mg/dL. The patient tested negative on IGRA, and he also tested negative for HIV antigens and antibodies, human T-cell leukemia virus type 1 antibodies, and anti-interferon-gamma autoantibodies. Gene panel tests for congenital immunodeficiency syndrome (*IL2RG*, *JAK3*, *IL7R*, *RAG1*, *RAG2*, *DCLRE1C*, *ADA*, *PNP*, *ZAP70*, *LIG4*, *NHEJ1*, *and TBX1*) and Mendelian susceptibility to mycobacterial diseases (*IL12RB1*, *IL12B*, *IL12RB2*, *IL23R*, *IFNGR1*, *IFNGR2*, *STAT1*, *CYBB*, *IRF8*, *TYK2*, *RORC*, *JAK1*, *IKBKG*, *GATA2*) were also negative.

Treatment for *M. genavense* was initiated with clarithromycin (CAM), rifampicin (RFP), ethambutol (EB), and amikacin (AMK). The patient’s fever temporarily improved, and the PSL dose was gradually tapered from 12.5 mg/day. One month after switching to oral therapy with CAM, RFP, EB, and levofloxacin, and 2 weeks after starting AMK treatment, the patient’s fever and pancytopenia recurred. The fever resolved immediately after the re-administration of AMK, and the pancytopenia improved. After 6 months of treatment, a treatment-resistant fever over 40 °C recurred, which persisted even with treatment with AMK combined with azithromycin, rifabutin, sitafloxacin, and EB. The patient also had pancytopenia, with a WBC count of 1100 /µL (lymphocytes, 27.0%; CD4 count, 53/µL). The Hb level was 7.8 g/dL, and the PLT count was 79,000/μL. PSL was increased from 7.5 mg/day to 30 mg/day (0.5 mg/kg/day), and the fever rapidly resolved. The pancytopenia improved, although the low CD4 count persisted. Five months later, hematology showed a WBC count of 3100/µL, with a lymphocyte count of 31.4% and a CD4 count of 212/µL. The hemoglobin level was 10.4 g/dL, and the PLT count was 124,000/μL.

We also reviewed Japanese cases of *M. genavense* infection. We used the Ichushi-Web, a search system specifically designed for Japanese literature, and Google, Google Scholar, PubMed, and reference lists to search for both Japanese and English language reports. The selection criteria for our literature search encompassed keywords “*Mycobacterium genavense*” or *“M. genavense*” in conjunction with “Japanese” or “Japan” for both Japanese and English language reports. We did not establish a specific time frame for the search period. A total of 17 Japanese cases of *M. genavense* infection, including the present case were identified (Table 1 and Table 2). All 17 patients were male, 10 (59%) were HIV-positive, and 7 (41%) were HIV-negative. Furthermore, 12 (71%) patients had a decreased CD4 count. The most common clinical manifestations were fever (13/17, 76%) and lymphadenopathy (13/17, 76%). In all cases, the diagnosis was confirmed by genetic analysis. The primary antibiotics used were macrolides (100%), ethambutol (100%), and rifamycin (16/17, 94%).

## 3. Discussion

We present here a case of a Japanese man who developed *M. genavense* infection during sarcoidosis treatment. Acid-fast bacteria detected by Ziehl–Neelsen staining were observed in the histopathological examination of a bone marrow biopsy sample; however, a definitive diagnosis could not be obtained through routine clinical testing. The diagnosis was confirmed using 16S rRNA sequencing. The patient tested negative for HIV and anti-interferon-gamma antibodies and underwent genetic testing for congenital immunodeficiency disorders. Treatment was initiated with a combination of multiple antibiotics, including aminoglycosides and macrolides; however, the patient developed resistance and experienced a persistently high fever and pancytopenia. Ultimately, a favorable response was achieved with steroid therapy. To gather evidence regarding epidemiology and response to treatment, and explore the characteristics of this rare disease, we also conducted a review of 17 Japanese cases of *M. genavense* infection.

Although most patients with *M. genavense* infection have HIV infection, *M. genavense* infection can occur in the absence of HIV infection. In the meta-analysis by Wetzstein et al. [19], 52 of the 223 (23.3%) individuals were HIV-negative, but most of them were also immunocompromised with conditions such as organ transplantation (15 cases, 6.7%), systemic lupus erythematosus (4 cases, 1.7%), and sarcoidosis (4 cases, 1.7%).

Notably, all 17 Japanese patients with *M. genavense* infection were male. In the meta-analysis by Wetzstein et al. [19], 171 of the 223 (76.7%) patients were HIV-positive, and 79.8% were male. Data from the National Institute of Infectious Diseases in Japan showed that 20,640 of 23,231 cases of HIV infection (88.7%) reported in 2021 were in males (https://www.niid.go.jp/niid/ja/aids-m/aids-iasrtpc/11555-512t.html). The higher proportion of males among patients with *M. genavense* infection in Japan can be attributed to a higher proportion of males among individuals with HIV infection. However, seven (41%) of the patients in our review were HIV-negative. It is unknown why all seven HIV-negative individuals were male. In an observational study from the Netherlands, 7 out of 9 (78%) HIV-negative patients with *M. genavense* infection were male [20]. Collectively, these results suggest that men have a higher risk of *M. genavense* infection than women, regardless of their HIV status, although further studies are required to confirm this finding.

A decrease in CD4 count is an important mechanism underlying infection, and cases of *M. genavense* infection have also been reported in individuals with idiopathic CD4 lymphocytopenia [21]. Among the 223 patients, 171 (76.7%) patients had HIV, and the median CD4 count was only 16/μL [19]. In addition, HIV-negative patients had low CD4 counts (median: 150/μL). In this literature review, 10 of the 17 patients had HIV (58.8%), and all these patients had CD4 counts less than 30/μL. Another two non-HIV patients (11.7%) including the present patient also showed lower CD4 counts (88 and 298/μL) [9]. Aside from HIV, immunosuppressant use in organ transplantation and various autoimmune diseases are also associated with lower CD4 counts. Thus, exploring the possibility of diseases associated with IL-12 receptor deficiencies and impairments of the interferon-gamma pathway might be important [13,20,22]. Among the 17 Japanese patients in this review, 3 (17.6%), 2 (11.8%), and 1 (5.9%) patient had collagen disease, was positive for anti-IFN-γ antibodies and had hypogammaglobulinemia. The possibility of idiopathic CD4 lymphocytopenia is also possible in the present case owing to the sustained low CD4 counts [23]. However, the possibility of bone marrow dysfunction resulting from the bone marrow infection itself should also be considered. The exact cause of the infection remains unknown.

*M. genavense* is difficult to culture using common mycobacterial culture methods. Diagnosis is challenging in nearly all cases and is confirmed only using molecular techniques such as 16S rRNA, rpoB, or hsp65 gene analysis [6,19,21]. In this literature review, all 17 Japanese patients had a positive smear, which led to the diagnosis. However, successful culture was achieved in only 7 patients (41%), and identification was subsequently confirmed through genetic analysis in all patients.

Available in vitro susceptibility data are limited [6]. However, existing data suggest that most isolates of *M. genavense* are susceptible to aminoglycosides, macrolides, quinolones, and rifamycins. A regimen containing macrolides has been demonstrated to be more effective than a regimen without macrolides. In the meta-analysis of 223 patients, logistic regression analysis revealed a significant association between macrolide administration and reduced occurrence of lethal events (OR: 0.162) [19]. In the present review, macrolides were used in all patients, and rifamycin was used in 94%. Quinolones and aminoglycosides were used only in 24% and 18% of cases, respectively. According to the American Thoracic Society guidelines, quinolones and aminoglycosides might be used more frequently in Japan than in other countries [6]. In the current case, aminoglycosides, macrolides, and rifamycins were administered as the initial treatment for *M. genavense*, and quinolones were added to achieve a five-drug regimen. However, the patient remained treatment resistant with prolonged fever and pancytopenia. The initial diagnosis was sarcoidosis 4 years previously, and steroid therapy was gradually tapered following the diagnosis of *M. genavense*. However, when the steroid dose was increased owing to treatment resistance observed with the five-drug regimen, the patient rapidly improved. Although steroid use should not be taken lightly owing to its side effects, apart from our report, there have been a few reports of the successful use of steroid combination therapy in cases of *M. genavense* infection in which antibiotic treatment alone failed to control the condition [7,24]. Further investigation is needed to validate this approach.

The initial diagnosis of sarcoidosis was based on the detection of granulomas in the lungs, skin, and bone marrow; increased lymphocyte count; the CD4/8 ratio in the BALF; and serum ACE levels. Multiple acid-fast bacilli smear examinations and culture tests were performed, and PCR tests for *M. tuberculosis* and MAC were also negative. However, anti-GPL-core IgA antibodies were also present at the initial diagnosis of sarcoidosis. The cause of sarcoidosis remains largely unknown, although mycobacteria have been reported as a contributing factor [25]. A case–control study of sarcoidosis that included 736 patients with sarcoidosis showed that organ involvement in sarcoidosis primarily affects the lungs (95%), skin (15.9%), lymph nodes (15.2%), eyes (11.8%), and liver (11.5%) [26]. Bone marrow involvement in sarcoidosis is relatively rare, occurring in only 3.9% of cases. In contrast, among the 223 patients with *M. genavense* infections, bone marrow was the most common organ affected, seen in 76 individuals (34%) [19]. Considering the rarity of bone marrow involvement in sarcoidosis, and the possibility of cross-immunity to *M. genavense* and MAC, it is possible that the initial diagnosis of sarcoidosis was a sarcoidosis-like condition caused by *M. genavense* infection. Although detecting *M. genavense* is quite challenging, the presence of granulomas in the bone marrow and anti-GPL-core IgA antibody positivity could signal the need to consider genetic testing for *M. genavense*.

## 4. Conclusions

We reviewed 17 Japanese cases of *M. genavense* infection, including the present case. We found that all 17 cases were male, and seven (41.2%) cases were HIV-negative, suggesting that males have a higher risk of *M. genavense* infection than female patients, regardless of their HIV status. Although the present case was HIV-negative, the CD4 count was low. In the search for the causes of *M. genavense* infection, the possibility of CD4 lymphocytopenia should be considered, not limited to HIV. The presence of granulomas in the bone marrow and the positive anti-GPL-core IgA antibodies may signal to consider genetic testing for *M. genavense*. Given that drug treatment with macrolides did not resolve the fever and pancytopenia, the steroid dose was increased, which resulted in a rapid improvement. Cases have been reported where despite administering adequate antimicrobial regimens for *M. genavense* infection, improvement was not achieved and steroids were ultimately required. Although further studies are needed, it is worth considering steroids in patients who do not respond adequately to antibiotic therapy. Our case report and Japanese case review contribute to a deeper understanding of this rare disease.

## Figures and Tables

**Figure 1 microorganisms-11-02145-f001:**
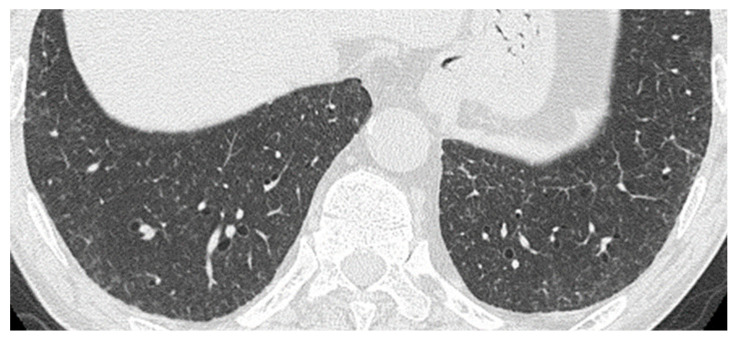
High-resolution computed tomography (HRCT) of the chest showing reticular shadows and thickening of the intralobular septa in the lower zones of both lungs at the time of diagnosis of sarcoidosis, 4 years before the diagnosis of *Mycobacterium genavense* infection.

**Figure 2 microorganisms-11-02145-f002:**
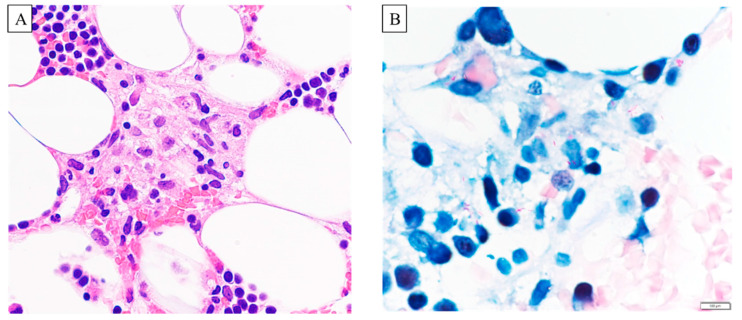
Pathological findings. (**A**): Hematoxylin and eosin (H&E) staining. (**B**): Ziehl–Neelsen staining. Pathological specimens obtained by bone marrow aspiration showing a noncaseating granuloma (**A**) and positive Ziehl–Neelsen staining (**B**). Scale bar: 100 µm.

**Table 1 microorganisms-11-02145-t001:** Details of 17 Japanese cases of *M. genavense* infection in the relevant English and Japanese literature.

Author, Publication Year	Age	Sex	HIV	Symptom	S ^1^	C ^1^	G ^1^	Detecting Samples	CD4 (/µL)	Treatment	Reference Number
Furunishi et al., 2006	43	M	(+)	Fever	(+)	(+)	(+)	Airway sample	7	RFP, EB, CAM, INH	[7] ^2^
Saito et al., 2006; Uchino et al., 2011	45	M	(−)	Fever, lymphadenopathy, hepatomegaly	(+)	(+)	(+)	Lymph node	N/A	RFP, EB, CAM, SM	[7,8]
Miyoshi et al., 2010	15	M	(−)	Abdominal pain, lymphadenopathy	(+)	(−	(+)	Digestive tract tissue, lymph node	298	RFP, EB, CAM	[9]
Niino et al., 2011	26	M	(+)	Fever, lymphadenopathy	(+)	(−)	(+)	Lymph node	25	RFP, EB, CAM, MFLX	[7,10] ^2^
Abe et al., 2013	23	M	(+)	Fever, cough	(+)	(+)	(+)	Digestive tract tissue, airway sample, blood	11	EB, AZM, LVFX	[11]
Koizumi et al., 2013	58	M	(+)	Lymphadenopathy	(+)	(+)	(+)	Digestive tract tissue, lymph node, blood	17	RFB, EB, CAM	[7] ^2^
Hoshina et al., 2014	44	M	(+)	Fever, lymphadenopathy	(+)	(−)	(+)	Lymph node	24	RFB, EB, CAM	[7] ^2^
Ogawa et al., 2015	44	M	(+)	Fever, lymphadenopathy	(+)	(−)	(+)	Bone marrow, lymph node	22	RFB, EB, CAM, LVFX	[7] ^2^
Tanaka et al., 2016	38	M	(+)	Fever, abdominal pain, lymphadenopathy	(+)	(−)	(+)	Bone marrow, lymph node, stool	25	RFP, EB, CAM, LVFX	[12]
Asakura et al., 2017	66	M	(−)	Abdominal pain, lymphadenopathy	(+)	(−)	(+)	Lymph node	Normal	RFP, EB, CAM, AMK	[13]
Yamashita et al., 2017	30s	M	(+)	Fever, abdominal pain, lymphadenopathy	(+)	(+)	(+)	Airway sample, stool, bone marrow, blood	10	RFB, EB, CAM	[14]
Hosoda et al., 2019	48	M	(+)	Fever, lymphadenopathy	(+)	(−)	(+)	Lymph node	3	RFB, EB, AZM, INH, LVFX, AMK	[15]
Oka et al., 2020	69	M	(−)	Fever, back pain	(+)	(+)	(+)	Bone marrow, blood	Normal	RFP, EB, CAM	[4]
Hosoda et al., 2020	73	M	(−)	Hemoptysis, lymphadenopathy	(+)	(−)	(+)	Airway sample	N/A	RFP, EB, CAM	[16]
Ito et al., 2020	53	M	(−)	Fever, lymphadenopathy, weight loss	(+)	(−)	(+)	Lymph node	678	RFP, EB, CAM	[17]
Murata et al., 2021	40	M	(+)	Fever, lymphadenopathy	(+)	(+)	(+)	Lymph node, blood	4	RFB, EB, CAM, INH	[18]
Present case (Ogata et al.), 2023	69	M	(−)	Fever	(+)	(−)	(+)	Bone marrow	88	RFP, EB, CAM, AMK	

The superscript numeral (^1^) indicates whether *M. genavense* is successfully detected (+) or not (−). This analysis is performed using smear (S), culture (C), and gene analysis (G) methods. The superscript numeral (^2^) indicates reference number 7 is a Japanese case report and review of Japanese previous cases. Abbreviations: AMK, amikacin; AZM, azithromycin; CAM, clarithromycin; EB, ethambutol; HIV, human immunodeficiency virus; INH, isoniazid; LVFX, levofloxacin; M, male; MFLX, moxifloxacin; N/A, not available; RFB, rifabutin; RFP, rifampicin; SM, streptomycin.

**Table 2 microorganisms-11-02145-t002:** Characteristics of 17 Japanese cases of *M. genavense* infection reported in the English and Japanese literature.

Age	Median: 44 years	Range: 15–73 years
	Number	Percentage
**Sex**		
Male	17	100%
Female	0	0%
**Underlying Conditions**		
HIV	10	59%
non-HIV	7	41%
Collagen disease	3	18%
Positive for anti-IFN-γ antibodies	2	12%
Hypogammaglobulinemia	1	6%
Possibility of idiopathic CD4 lymphocytopenia	1	6%
Unknown	2	12%
**CD4 (cells/μL)**		
1–100	11	65%
100–500	1	6%
501+	3	18%
Unknown	2	12%
**Clinical manifestations**		
Fever	13	76%
Lymphadenopathy	13	76%
Abdominal pain	4	24%
**Affected organs**		
Lymph nodes	11	65%
Bone marrow	5	29%
Gastrointestinal tissue	5	29%
Respiratory system	4	24%
Blood	4	24%
**Tests**		
Smear	17	100%
Culture	7	41%
Genetic analysis	17	100%
**Treatment Medication**		
Macrolide	17	100%
Clarithromycin	15	88%
Azithromycin	2	12%
Ethambutol	17	100%
Rifamycin	16	94%
Rifampicin	10	59%
Rifabutin	6	35%
Quinolone	4	24%
Levofloxacin	4	24%
Moxifloxacin	1	6%
Aminoglycoside	3	18%
Amikacin	3	18%
Streptomycin	1	6%
Isoniazid	3	18%

## Data Availability

The datasets for the current case and reviews are available from the corresponding author upon reasonable request.

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
