# Peer review of "Disseminated Mycobacterium genavense Infection Mimicking Sarcoidosis: A Case Report and Review of Literature on Japanese Patients"

_microorganisms, 2023, doi:10.3390/microorganisms11092145_

Round 1

Reviewer 1 Report

This case report and literature review is well presented. I just have two minor suggestions.

1. Please describe the literature search methods, such as key words, and period.

2. Please delete the "resistance" in the abstract and text. It could be misunderstand as "antibiotic resistance"

Reviewer 2 Report

Ogata et al. present a case study describing misdiagnosis of M. genavense infection as sarcoidosis and provide a literature review if 17 cases (including the current case).

This is an important case to present because of the clinical similarities between both diseases and the importance of initiating the proper treatment early in the course of the disease.

The report is well prepared and contains all the crucial components as a publication. I have the following comments, however. 

Minor comment:

Line 32; add ‘to’ in “challenging identify”.

Other comments:

(1) It would be more helpful for the reader if the authors elaborate on sarcoidosis disease in the introduction. In terms of causes, clinical aspects, and treatment.

(2) The authors write (lines 228-229) that “it is reasonable to assume that this is a case of M. genavense infection at the time of diagnosis of sarcoidosis”

This is not a reasonable assumption because the work presented in this report, in terms of clinical evaluation and diagnosis, was performed 4 years after the initial diagnosis of sarcoidosis. One can not exclude the possibility that M. genavense infection occurred some time after the onset of sarcoidosis.

(3) In Table 1, the authors summarize the details of 17 Japanese cases of M. genavense infection. 7 of these reference one report for Ogawa et al., 2015. Reference format in this submission used numbers and not Last name of authors. The case references in Table 1 should also use reference numbers in another column or replace the first column of author’s last name with the reference number identified in the list of references.

Another point here is referencing 7 cases to this one author. Were the seven cases detailed for 7 different patients in the same report? The reference specifies a case report, not 7 cases. The authors reviewed the literature for 6 cases in that report. Does that information need to be repeated here? The authors need to explain the importance of including this information.

(4) If the authors are convinced that there was a misdiagnosis of sarcoidosis initially, they should provide a differential diagnosis to separate both diseases.

(5) The authors should include the selection criteria used in their literature search. What key words were used, the year, etc.

English language is of good quality.

Reviewer 3 Report

The works is a case report on a patient with a diagnosis of sarcoidosis that was deeply recognized as infection by Mycobacterium gevanense by 16s sequencing.

Some points need to be revised to respect section of the manuscript and clarify some points:

-        Avoid a list of parameters and later a list of their value. It’s easy for the reader to organize tha parameter and it’s value, especially if the list is long (ie. lines 50-51, 56-58)

-        The discussion reports data and not discussion of data. Please, include all data in results section and use references to discuss interesting data in discussion section. Update bibliography.

-        At line 186 is reported “review”. Please, correct the statament.

-        Clarify lined 210-211 regards the therapy.

-        The conclusion section reports again all the finding of the work, from the beginning. Please, revise the section.

Moderate editing of English language required

Round 2

Reviewer 3 Report

The revision process strongly enhance the quality of the work.